# Model Based Simulation and Genetic Algorithm Based Optimisation of Spiral Wound Membrane RO Process for Improved Dimethylphenol Rejection from Wastewater

**DOI:** 10.3390/membranes11080595

**Published:** 2021-08-04

**Authors:** Mudhar A. Al-Obaidi, Alejandro Ruiz-García, Ghanim Hassan, Jian-Ping Li, Chakib Kara-Zaïtri, Ignacio Nuez, Iqbal M. Mujtaba

**Affiliations:** 1Technical Institute of Baquba, Middle Technical University, Baqubah 00964, Iraq; dr.mudhar.alaubedy@mtu.edu.iq; 2Department of Electronic Engineering and Automation, University of Las Palmas de Gran Canaria, 35017 Las Palmas de Gran Canaria, Spain; ignacio.nuez@ulpgc.es; 3Institute of Technology, Middle Technical University, Baghdad 10074, Iraq; dr.ghanim@mtu.edu.iq; 4Department of Chemical Engineering, Faculty of Engineering and Informatics, University of Bradford, Bradford BD7 1DP, West Yorkshire, UK; J.P.Li@bradford.ac.uk (J.-P.L.); C.Karazaitri@bradford.ac.uk (C.K.-Z.); I.M.Mujtaba@bradford.ac.uk (I.M.M.)

**Keywords:** wastewater treatment, spiral wound reverse osmosis, modelling, species conserving genetic algorithm optimisation, dimethylphenol removal, energy consumption

## Abstract

Reverse Osmosis (RO) has already proved its worth as an efficient treatment method in chemical and environmental engineering applications. Various successful RO attempts for the rejection of organic and highly toxic pollutants from wastewater can be found in the literature over the last decade. Dimethylphenol is classified as a high-toxic organic compound found ubiquitously in wastewater. It poses a real threat to humans and the environment even at low concentration. In this paper, a model based framework was developed for the simulation and optimisation of RO process for the removal of dimethylphenol from wastewater. We incorporated our earlier developed and validated process model into the Species Conserving Genetic Algorithm (SCGA) based optimisation framework to optimise the design and operational parameters of the process. To provide a deeper insight of the process to the readers, the influences of membrane design parameters on dimethylphenol rejection, water recovery rate and the level of specific energy consumption of the process for two different sets of operating conditions are presented first which were achieved via simulation. The membrane parameters taken into consideration include membrane length, width and feed channel height. Finally, a multi-objective function is presented to optimise the membrane design parameters, dimethylphenol rejection and required energy consumption. Simulation results affirmed insignificant and significant impacts of membrane length and width on dimethylphenol rejection and specific energy consumption, respectively. However, these performance indicators are negatively influenced due to increasing the feed channel height. On the other hand, optimisation results generated an optimum removal of dimethylphenol at reduced specific energy consumption for a wide sets of inlet conditions. More importantly, the dimethylphenol rejection increased by around 2.51% to 98.72% compared to ordinary RO module measurements with a saving of around 20.6% of specific energy consumption.

## 1. Introduction

The modern industrial world continues to produce a wide range of harmful organic and non-organic compounds. These pollutants are usually disposed of into a variety of water sources, which in turn have a serious impact on the biological ecosystem [1,2]. This study focuses on phenolic compounds, and especially dimethylphenol, due to their existence in several industrial effluents such as those from refineries and petrochemical plants [3]. Dimethylphenol contains a stable benzene ring, which increases its resistance to biological decomposition and therefore lingers in the environment for a long period of time [4]. Moreover, the hydrophobicity property of phenolic compounds yields the formation of toxicological organic and free radical species, which are very harmful [5]. Several health agencies rate phenol and phenol derivatives as toxic compounds (even at low concentrations). Dimethylphenol is therefore tightly controlled by legislation due to its carcinogenic properties [6]. For example, the ATSDR (Agency of Toxic Substances and Disease Registry) [7] has restricted dimethylphenol concentration in surface water to 0.05 ppm.

UV/H_2_O_2_ technology has been used as the prominent treatment process for the elimination of phenol and its derivatives from wastewater. However, this technology consumes a significant amount of energy and with increased carbon concentration of the reused water [8].

Reverse Osmosis (RO) membrane is a well-known water treatment method, which has been used extensively in seawater desalination [9]. The RO process has confirmed its efficiency in terms of low cost of operation and low energy consumption compared to thermal process methods [10]. The use of RO has therefore been extended to treat wastewater from various other industries [11]. For example, RO was magnificently used to eliminate heavy metals such as copper, nickel, acrylonitrile, sulphate, ammonium, cyanide and sodium [12,13]. Generally, the RO process and especially spiral wound membrane method remains the most promising treatment method for the removal of several highly toxic compounds. Whilst various published studies have confirmed RO’s efficiency for treating secondary effluents at low cost, the challenge for enhancing its performance for rejecting toxic compounds from wastewater is yet to be explored fully [14,15].

The efficiency of a spiral wound membrane module to remove such harmful compounds is dependent on the membrane type, design parameters and control variables such as the feed pressure, flow rate, concentration and temperature. Several attempts can be found in the literature for improving the efficiency of seawater RO process using optimisation methods [16]. However, only a few of such attempts have been carried out to optimise the membrane design parameters for wastewater treatment.

Boudinar et al. [17] enhanced the efficiency of a ROGA-4160HR spiral wound membrane module used for seawater desalination using a geometric optimisation for one set of input conditions. Sharifanfar et al. [18] assessed the influence of channel height on the permeate flux of a microfiltration membrane for a pomegranate juice clarification process, and concluded that the feed channel height had an impact on the permeate volume. Karabelas [19] studied the effect of membrane sheet dimensions of a spiral wound membrane module used to desalinate seawater based on a fixed effective membrane area efficiency. He used an optimisation methodology based on the geometric characteristics of feed-side spacers. Gu et al. [20] explored the influence of the winding geometry of a spiral wound membrane RO module used for seawater desalination on the total process performance and energy consumption. The parameters studied included the membrane dimensions, number of membrane leaves, centre pipe radii, height of feed and permeate channels. Ruiz-García and de la Nuez Pestana [21] considered the impact of different feed spacer geometries on three different full-scale spiral wound membrane modules. They analysed the performance of membrane elements for wide ranges of feed concentration, pressure and flowrate. Toh et al. [22] studied the 3D feed spacer geometries of a spiral wound membrane RO module with various degrees of “floating” characteristics via the implementation of Computational Fluid Dynamics (CFD) simulations to explore the mechanisms that result in shear stress and mass transfer improvement. Luo et al. [23] presented a hybrid framework of CFD model to explore the optimal design of feed spacer in a non-woven spiral wound membrane RO module and analyse the influence of industrial operating conditions on the performance of brackish water RO process.

This research focuses on exploring the removal of dimethylphenol from industrial effluents using a spiral wound RO membrane module. Al-Obaidi et al. [24] studied the effect of membrane dimensions including membrane length, width, and feed channel height on the retention of dimethylphenol from synthesised wastewater and the total consumed energy of a single spiral wound membrane module. They used a simulation model based on the solution-diffusion principle. However, this was carried out for only one set of inlet parameters of 6.548 × 10^−3^ kmol/m^3^, 13.58 atm, 2.583 × 10^−4^ m^3^/s, and 31.5 °C of feed concentration, pressure, flow rate, and temperature, respectively. A gPROMS software optimisation tool has been used in this study to optimise the removal of dimethylphenol from wastewater by considering the membrane design parameters as the decision variables. However, the results of Al-Obaidi et al. [25], which were obtained using gPROMS software, were based on a discrete solution of a single objective function. Additionally, gPROMS software cannot provide a set of alternative solutions that trade various objectives against each other. This is to say that it would be interesting to explore the results of a multi objective optimisation approach. This should provide a set of cooperative optimal solutions (alternatives) as confirmed by Savic [26]. The author compared the feasibility of single and multi-objective optimisation methods applied in water distribution design and affirmed that a multi-objective function optimisation-based model can be a useful tool for formulating alternative objectives especially for such systems with high uncertainty, and which can readily be used for exploring trade-off opportunities. This prompted the use of Genetic Algorithms (GA) as an evolutionary computation technique [27] for finding one global solution.

To improve the performance of GAs in identifying global solutions, several methods can be used to solve multimodal problems. They include crowding, fitness share, clearing, multi-national GA and species conserving [27,28]. More specifically, the Species Conserving Genetic Algorithm (SCGA) can generate several solutions of complex optimisation problems [29]. For this reason, SCGA has been selected for solving the proposed optimisation problem.

### The Use of Genetic Algorithms for Developing a Global Optimisation Solution

Traditional Genetic Algorithm (GA) based optimisation methods have been implemented in several applications including wastewater treatment. For example, Al-Obaidi et al. [30] applied traditional GA to find the optimal chlorophenol rejection from wastewater using a single spiral wound RO membrane module. Al-Obaidi et al. [31] researched the best configuration of multistage RO processes based on permeate-reprocessing to reject N-nitrosodimethylamine (NDMA) from industrial effluents. To the best of the authors’ knowledge, the effect of a wide set of operating parameters on the removal of dimethylphenol from wastewater via a simulation of a single spiral wound RO membrane module has not been yet addressed in the literature.

This research attempts to use, for the first time, the Species Conserving Genetic Algorithm (SCGA) for significantly improving the RO process performance for variable inlet conditions associated with variable membrane design parameters that will yield a higher dimethylphenol rejection from wastewater at low energy consumption. The main output of this work is the generation of multiple optimal solutions for any set of operating data [28]. The net effect of this is the ability to select the most suitable solution based on process requirements.

This paper begins with an overview of an earlier mathematical model developed by the same authors [24]. This was successfully applied to simulate the transport phenomenon of permeate and solute via the membrane texture of a spiral wound membrane module. A comprehensive analysis of the validated results against a wide range of experimental data from the literature is provided. The effect of membrane design parameters, such as membrane length, width, and feed channel height, are then assessed in respect of the rejection of dimethylphenol from industrial wastewater. This is used for two different sets of variable inlet conditions, as well as the required energy consumption for given operating conditions. Finally, the process model is developed in the gPROMS software, and the multi-objective optimisation problem is implemented using the species conserving genetic algorithm written in C++. This yields the most economical membrane design parameters, thus providing the highest dimethylphenol rejection at lowest energy consumption. In this regard, it is important to clarify that the implementation of the species conserving genetic algorithm based a model developed to allocate the optimal values of membrane dimensions of a single spiral would RO process that would maximise the dimethylphenol removal from wastewater at the lowest specific energy consumption has not been addressed yet. Therefore, this study attempts to resolve this challenge.

## 2. Materials and Methods

### 2.1. Modelling of a Spiral Wound Membrane Module of RO Process

Sundaramoorthy et al. [32] developed an analytical model based on the solution-diffusion model, which has been originally deployed by Srinivasan et al. [33] to investigate the performance of a single spiral wound RO membrane module for the removal of dimethylphenol from synthesised wastewater.

The model used in this study was first established by Al-Obaidi et al. [24], who used it to distinguish the transport phenomenon and allow the consideration and estimation of the required energy consumption. The model assumptions are as follows:(a)The solution-diffusion model characterises the solvent and solute fluxes.(b)The film theory identifies the membrane wall concentration.(c)Darcy’s law quantifies the pressure drop along the feed side of the module.(d)There is a fixed 1 atm pressure at the permeate side.(e)Membrane transport parameters are fixed, i.e., solute, and solvent transport parameters and constant friction parameter.(f)No temperature difference throughout the operation.(g)Constant high-pressure pump (HPP) efficiency of 80%.(h)The influence of pH variation has not been considered.

Table A1 of Appendix A presents the model and physical property equations for a single spiral wound membrane RO module to simulate and optimise the rejection of dimethylphenol from its aqueous solutions. As in many references, the water relationships anticipated by Koroneos et al. [34] have been used to calculate the physical properties of low concentration dimethylphenol aqueous solutions.

The nonlinear algebraic correlations of the proposed model (given in Table A1 of Appendix A) can be presented in the following compact formula.
f(*x*, *u*, *v*) = 0(1)

In this form, *x*, *u*, and *v* represent the set of all algebraic control variables, decision variables, and constant parameters, respectively.

The research outlined in this paper has included the calculation of specific energy consumption. This would be considered as a significant improvement made in this study since both [32,33] did not include the energy consumption parameter in their model. Thus, the most interesting question of this research is how to attain a higher rejection rate at a lower energy consumption. However, it is firstly important to validate the model of Al-Obaidi et al. [24] (provided in Appendix A) against the experimental data of [33] to quantify its consistency.

### 2.2. Experimental Setup 

For the convenience of the readers, we highlight the experimental work of Srinivasan et al. [33] here briefly. They conducted extensive experiments to assess the feasibility of the RO process to reject dimethylphenol from synthesised dilute wastewater of different concentrations. Specifically, they used a single spiral wound membrane module of thin film composite RO membrane. The features of the membrane module are chosen the same as from [33] and are given in Table 1. The applied feed concentration of dimethylphenol varies between 0.819 × 10^−3^ and 6.548 × 10^−3^ kmol/m^3^. Additionally, the operating feed flow, pressure, and temperature were selected between 2.166 × 10^−4^ to 2.583 × 10^−4^ m^3^/s, 5.83 to 13.58 atm, and 29 to 32.5 °C, respectively. The water and dimethylphenol transport parameters throughout the membrane and friction factor (*A*_w_, *B*_s_ and *b*) are also given in Table 1.

Figure 1 depicts a detailed diagram of the corresponding experimental setup of a single spiral wound membrane RO module with the corresponding items. Figure 2 displays a 3D representation diagram of the flat sheet membrane. The inlet wastewater of aqueous solution of dimethylphenol splits into two streams of permeate (collected at the permeate channel) and retentate of high concentration dimethylphenol. This is due to supplying a higher pressure than the osmotic pressure, which helps high quality water to penetrate through the membrane pores. The experimental data of [33] will be used in the next section to validate the new model, which equations are shown in Table A1 of Appendix A.

### 2.3. Model Validation by Al-Obaidi et al.

The experimental data of Srinivasan et al. [33] were used to validate the model developed (shown in Table A1 of Appendix A) by Al-Obaidi et al. [24], which is used in this study for simulation and optimisation. Table A2 of Appendix A shows such data of [33] and the model calculations for each set of inlet conditions. The results clearly show that insignificant percentage errors exist between the model calculations and experimental data.

### 2.4. Optimisation Methodology

#### 2.4.1. Problem Description

In this section, the optimum dimethylphenol removal and the minimum energy consumption are simultaneously investigated via the optimisation of membrane dimensions, including the length, width, and feed channel height of a single membrane module of RO process. The optimisation study is carried out using the SCGA platform, based on the model correlations and the restricted upper and lower bounds of membrane module design parameters. The optimisation considered is based on a fixed membrane area of 7.84 m². This constraint was chosen to meet the manufacturer specification of membrane area and technical requirements. The decision variables are selected between the upper and lower bounds and taken as 0.5–1 m of membrane length, 5–15.69 m of membrane width, and 5.93 × 10^−4^–1 × 10^−3^ m of feed channel height. Additionally, the model parameters of permeate and dimethylphenol transport parameters and friction factor (*A*_w_, *B*_s_ and *b*) are assumed constant (Table 1). Most importantly, the optimisation is carried out for several operating parameters of feed flow rate, concentration, pressure and temperature. These are used to investigate the appropriate operating conditions including the optimum membrane design parameters commensurate with the optimum efficiency of a single spiral wound membrane RO module.

The multi-objective function is targeted to simultaneously maximise the dimethylphenol removal and minimise the energy consumption. This is represented mathematically as follows:
Max and MinRej, EC, respectively*L*, *W*, *t*_f_
Subject to: Equality constraints: 
Process Model: f (*x*, *u*, *v*) = 0Inequality constraints: 0.5 ≤ *L* ≤ 1.05.0 ≤ *W* ≤ 15.695.93 × 10^−4^ ≤ *t*_f_ ≤ 1 × 10^−3^Equality end-point constraint: A = 7.84 m^2^

Note, Al-Obaidi et al. [24] used the same optimisation problem formulation but used the gPROMS Model Builder to solve the optimisation problem using Point Optimisation technique that considers the Nonlinear Programming problems (NLP). This method is mathematically comparable to solve an algebraic problem with bearing in mind neither minimising or maximising a nonlinear objective function exposed to equality and inequality nonlinear constraints of upper and lower bounds of process operation. The optimisation problem is therefore solved by controlling a set of optimisation continuous or discrete control variables. Thus, the suitable control variables can be estimated to fit the projected objective function. However, gPROMS Model Builder cannot simultaneously solve several objective function and therefore, the multi objective function is solved after running the optimisation for an individual objective function and incorporating the second objective function as a constraint.

In this work we used a different optimisation technique based on SCGA as described below that enables to solve multi objective functions in one run.

#### 2.4.2. Description of Species Conserving Genetic Algorithm (SCGA)

A species is an important term in SCGA [27] that represents a set of similar individuals. Species are identified from a population. Specifically, a species *s_i_* is dominated by its species seed *x*^*^, which has the greatest fitness value (objective value), if for everyone *y* ∈ *s_i_*
d(*x**,*y*) < *r_s_*,(2)
and
*f*(*y*) ≤ *f*(*x*),(3)
d(*,*) represents the distance between two individuals, and *r_s_* is the species radius. A possible species distribution in a 2-D space is illustrated in Figure 3. A species contains some individuals and is a part of the practicability section. However, some individuals may be connected to many species. The pseudo codes of SCGA [27] are described in Figure A1 of Appendix A. In this regard, *G*(*t*) signifies the population at time *t*, and *x* signifies the species set.

A population is dynamically divided into subgroups, called species. Each species seed is a possible solution. The concept of SCGA is to locate species and make them survive in the next generation. Three operators in SCGA are added into a traditional GA. This in turn represents the main differences existing between traditional GA and SCGA. The three SCGA operators, are more particularly discussed below.

Identifying species seeds: This operator was developed to explore all the possible species from the current population. Firstly, all the individuals are set as untreated. Then, a best untreated individual is chosen to be a species seed of a species. An individual will be marked as the member of the species if its distance to the species seed is smaller than the species radius, and will therefore be marked as “processed”. This practice is recurrent until all the individuals have been marked.Conserving species seeds: The selected species seed is imitated back to the population and will replace the nearest individual if it is better the individual. The goal of this process is to ensure that all the species can continue in the next generation.Identifying global solutions: This is achieved by choosing the top species from xs due to saving the best individual in a species in the set xs. A threshold *r_f_* (0 < *r_f_* ≤ 1) is used to find the global solutions. A species seed *x* is therefore treated as a solution, if:

*f*(*x*) ≤ *f*_max_ − (*f*_max_ − *f*_min_) *r*_f_,(4)

*f*_min_ and *f*_max_ are the worst and best objective values. The defaulting value of species radius is established as 1 in this research.

Figure A1 of Appendix A illustrates a typical genetic algorithm when the above three operators are removed. This includes three genetic operations: selection, crossover and mutation. After those operations, new individuals from the crossover and mutation will be evaluated. Most importantly, SCGA can provide multi optimal solutions for a single objective function. Therefore, the multi-objective functions presented in Section 2.4.1 should be calibrated to represent a single objective function. This is readily achieved by developing a new formula (Equation (4)) based on weighting factors to arrive at a single objective function derived from two objective functions commensurate with SCGA requirements.
*f*(*L*, *W*, *t*_f_ ) = *W*_1_ × *Rej* + *W*_2_ × (1/*EC*)(5)

*W*_1_ and *W*_2_ are the weighting parameters. Thus, the main aim of this optimisation is to maximise the objective function f in Equation (4). However, we assume that the two objectives are at the same level of importance. In other words, the maximum value *Rej* (overall rejection) is set as 1 and the maximum value of *EC* is about 3 for simplification purposes. Thus, let *W*_1_ to be 1, and *W*_2_ be 3, which denotes that both objectives are at an identical level of importance and have a similar involvement of the system objective.

## 3. Results and Discussions

### 3.1. Steady-State Simulation

In this section, the model presented in Table A1 of Appendix A and validated in Section 2.3 is implemented to carry out a simulation to forecast the effect of inlet parameters on dimethylphenol rejection from wastewater for a spiral wound membrane RO module. Firstly, the simulation indicates that the rejection parameter and total water recovery grow due to a rise in feed pressure at any fixed operating feed flow rate, concentration and temperature (for instance, Experiments 2 to 4, Table A2 of Appendix A). This is due to the water flux increase, which in turn is related to the increasing pressure supplied (Equation (A1), Table A1 of Appendix A). This reduces the permeate concentration of dimethylphenol despite insignificantly enhancing the removal of dimethylphenol. The consequence of this is (insignificant) reduction of the energy consumption as outlined in Equation (A29), where any increase of water production serves to limit the energy consumption despite the increase of the operating pressure. Statistically, rising the feed pressure from 9.71 to 13.58 atm at fixed other inlet conditions, would result in 0.07% decrease of energy consumption.

Secondly, simulating the process at fixed feed flow rate, pressure and temperature and increased feed concentration (for instance, Experiments 12 and 15, Table A2 of Appendix A) yields a significant decrease of total permeate recovery and an increase in the removal of dimethylphenol. This might be ascribed to the growth of the osmotic pressure because of the increasing feed concentration, which generally decreases the water flux. The simulation results of Experiments 12 and 15 (Table A2 of Appendix A) showed that the osmotic pressure has been increased from 0.84 to 1.065 atm, which corroborates the validity of the reason above. However, this increased dimethylphenol rejection can be elucidated by the fact that increasing bulk concentration is not necessarily comparable to the increase of permeate concentration, as this is due to the increasing operating concentration. It is argued therefore, that the rejection parameter increases, as outlined by Equation (A28) in Table A1 of Appendix A, due to the lifting of the feed concentration. More generally, any growth of feed concentration is associated with an increase of energy consumption.

Finally, the increase of feed flow rate at constant inlet parameters of operating pressure, concentration and temperature (for instance, Experiments 8 and 17, Table A2 of Appendix A) results in an insignificant rejection increase the but with a noticeable reduction of water recovery. This phenomenon can be ascribed to the reduced resident time of the fluid inside the feed channel, which itself is due to the increased feed flow rate. Such finding can be described by the high frictional pressure drop that reduces the water flux, and in turn increase the energy consumption required.

### 3.2. Influence of Membrane Design Parameters

The effect of membrane design parameters, which include membrane length, width and feed channel height as shown in Table 1 (as per manufacturer’s specification), is assessed in respect of dimethylphenol removal, water recovery, and the consumed energy at two particular sets of operating parameters. These were the same as those experimental data used by Srinivasan et al. [33] at the highest and lowest dimethylphenol rejections achieved. More specifically, the highest rejection of 97.3% is commensurate with 6.548 × 10^−3^ kmol/m^3^, 13.58 atm, 2.583 × 10^−4^ m^3^/s, and 31.5 °C of inlet concentration, pressure, flow rate and temperature, respectively. The lowest rejection of 90.2% is commensurate with 0.819 × 10^−3^ kmol/m^3^, 2.166 × 10^−4^ m^3^/s, 5.83 atm, and 32.5 °C, respectively. The next sections present the simulation results in detail at the two selected sets of operating conditions.

#### 3.2.1. Influence of Membrane Dimensions of Length and Width

The membrane dimensions of length and width are altered at fixed volume and membrane area. This is done to adjust the flow patterns of the fluid inside the feed channel and will be used to assess the extent of dimethylphenol removal, permeate recovery, and energy consumption. The geometrical amendment of the selected membrane (Ion Exchange, India) is achieved at the two selected operating parameters (provided in Section 3.2) of inlet concentration, pressure, flow rate and temperature.

Figure 4 and Figure 5 show the effect of variable membrane dimensions at the fixed membrane area and feed channel height of 7.84 m² and 0.8 × 10^−3^ m, respectively on dimethylphenol removal for the two sets of operating conditions. Figure 6 presents the effect of membrane width at fixed membrane area and feed channel height on the permeate recovery and energy consumption. A slight increase was noticed in dimethylphenol rejection as a result of increasing membrane length at fixed area and feed channel height (Figure 4 and Figure 5). Clearly, any decrease in membrane width would enhance dimethylphenol rejection. This is owing to a rise in the bulk velocity inside the membrane module, which in turn is due to the membrane width decreasing (Equation (A16), Table A1 of Appendix A) or the membrane length increasing at fixed membrane area. Both result in a reduction of the wall membrane concentration and concentration polarisation, which yield less solute flux and more dimethylphenol rejection.

Figure 6 confirms an increase in permeate recovery due to an increase in membrane width at fixed membrane area. This reduces the pressure drop, which rises the permeate flux through the membrane, and this applies to both sets of tested operating conditions. Similarly, the energy consumption required decreases due to the membrane width increasing—this is especially so for the second set of operating conditions (Figure 6). The reason for this is the increase of permeate recovery due to the increase in the membrane width. Another interesting point here is that running the process at the second set of operating conditions (the lowest rejection) can generate a higher permeate flux as a response of the membrane width variation compared to the first set of operating conditions (the highest rejection). Karabelas et al. [19] confirmed that using short sheets of membrane can potentially improve the recovery performance of low and high-pressure membranes.

#### 3.2.2. Influence of Feed Channel Height

Figure 7 shows the influence of feed channel height on dimethylphenol removal and energy consumption for the two sets of inlet conditions mentioned in Section 3.2. This simulation is carried out at the fixed membrane area of 7.84 m^2^ and variable module volume and fixed feed conditions. Interestingly, the increase of the feed channel height actually reduces dimethylphenol rejection but increases the energy consumption (Figure 7). The cause of this is a rise in the height of feed channel and the pressure drop. The decline in water flux through the membrane, thus dimethylphenol rejection, is the main reason for the rise in the consumption of energy (Equation (A29) in Table A1 of Appendix A). Seemingly, running the process using the second set of operating conditions significantly decreases dimethylphenol rejection compared with doing the same with the first set of operating conditions. This is due to lower water flux for the second set of operating conditions when the feed channel height increases. This results in a higher concentration of the pollutant (dimethylphenol) at the permeate channel. In contrast, the first set of operating conditions cause high operating pressure and therefore higher water flux despite feed channel height variation. These results are corroborated by Sablani et al. [36], who confirmed that the feed channel height of the spiral wound membrane module has a substantial influence on the performance indicators of the RO seawater desalination process.

The above results readily provide a clear motivation for an optimisation study for analysing the precise impact(s) of all membrane design parameters within the objective functions and operational constraints. The optimisation study is discussed in more detail in the next section.

### 3.3. Optimisation Results Based on a Species Conserving Genetic Algorithm (SCGA)

The optimal values of the membrane length, width and feed channel height (decision variables) obtained by SCGA are given in Table 2 for several selected control variables of inlet concentration, pressure, flow rate and temperature, respectively. Table 2 presents several optimal solutions for each set of operating parameters, while the best solution is highlighted (Bold). This is done by a simple comparison of the rejection and energy consumption obtained including the proposed solutions. However, the highlighted optimal solutions of different operating conditions show the consistency of these solutions obtained by SCGA compared to the experimental results of Srinivasan et al. [33]. In this regard, Al-Obaidi et al. [24] confirmed the optimum solution of the same membrane as 9.745 m, 0.805 m, and 5.93 × 10^−4^ m of membrane width, length and feed channel height, respectively using the gPROMS suite optimisation tool. The optimised solution of Al-Obaidi et al. [24] is quite close to the output of the SCGA solution 3 for the operating conditions under case 2, as presented in Table 2. However, SCGA has generated multiple optimised solutions compared to only one solution provided by the gPROMS optimisation tool. This is the main advantage of using SCGA for multi-objective optimisation compared to gPROMS, which handles a single objective function as part of the optimisation process.

Table 2 shows the original dimensions of the membrane selected by Srinivasan et al. [33] and a comparative SCGA analysis of dimethylphenol rejection and energy consumption results. Therefore, optimal solution 1 of the operating conditions (case 1) yields the best process performance results against the experimental data of [33]. In this respect, Table A2 of Appendix A shows the simulation results by means of the optimised solution 1 of the membrane design parameters. This in turn yields the optimised recovery, dimethylphenol rejection and the percentage of energy saving compared to the original membrane design parameters of Srinivasan et al. [33], for each set of operating parameters.

It can readily be seen from the new optimisation results that the proposed methodology can be used to achieve the maximum dimethylphenol rejection at the minimum energy consumption for all the inlet parameters of [33]. The corresponding energy saving varies between 0.79% to 20.66% based on the fundamental set of inlet parameters (Table A2 of Appendix A). In this regard, the minimum and maximum optimum energy consumptions are 1.24 and 1.695 kWh/m^3^, respectively, based on the fundamental set of inlet parameters. Additionally, the rejection of dimethylphenol has been increased between 2.51% to 5.87% to attain 98.72% and 98.14%, respectively, based on the fundamental set of inlet parameters (Table A2 of Appendix A. These results are comparable to the maximum energy saving of 19.2% reported by Al-Obaidi et al. [24] for the same membrane using gPROMS. Additionally, the new optimisation results readily show that the most economical performance is achieved with a specific intermediate spacer thickness (5.93 × 10^−4^ m) compared to the manufacturer specifications. An immediate and interesting outcome here is that wastewater usually comes with a low pollutant concentration, which means with a low possibility of fouling. The implementation of the optimised feed channel height of 5.93 × 10^−4^ m will therefore result in a much lower risk of membrane blogging.

It would therefore be safe to say that the new optimisation methodology of the membrane design parameters, which yielded improved pollutant rejection at lower energy consumption for a spiral wound membrane RO module, can readily be applied to any type of organic pollutant such as chlorophenol and phenol. Having said this, full details of the membrane transport coefficients of the water and pollutant should be known besides the physical properties.

## 4. Conclusions

Dimethylphenol compounds, found in several industrial effluents, are extremely resistant to any biological decomposition and can readily cause serious harm to humans and the environment. This is why dimethylphenol concentration in surface water has been limited to 0.05 ppm by health agencies. The aim of this research was to obtain an efficient method for removing this toxic compound from industrial wastewater using a single membrane RO module. This has been achieved by using a comprehensive simulation based model for analysing the influence of the membrane length, width and feed channel height (membrane design parameters) on the removal of dimethylphenol, total permeate recovery and energy consumption. Firstly, the consistency of the model developed has been tested against experimental data from the literature. Simulation results confirmed that the geometric parameters of membrane length and width have a minor impact on the rejection rate on one hand, and a marked impact on energy consumption on the other. Ever-increasing the feed channel height has a negative influence on both dimethylphenol rejection and energy consumption. Finally, the model was used to carry out a multi-objective optimisation for the membrane design parameters using a species conserving genetic algorithm. The results of the optimisation analysis yielded an optimum removal of dimethylphenol at reduced energy consumption (objective functions) of the RO process. Specifically, the optimisation showed a higher dimethylphenol rejection (around 5.8%) at lower energy consumption (around 20.6%) when compared to ordinary RO module measurements.

## Figures and Tables

**Figure 1 membranes-11-00595-f001:**
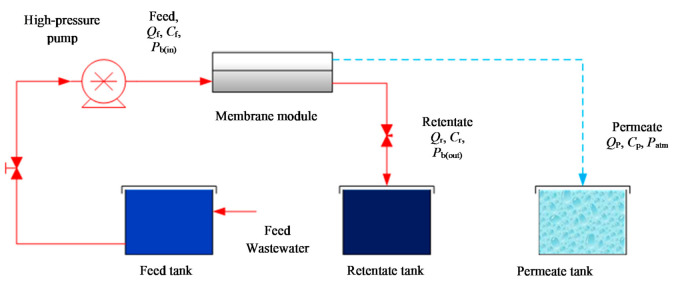
A detailed diagram of a single spiral wound membrane module of RO process.

**Figure 2 membranes-11-00595-f002:**
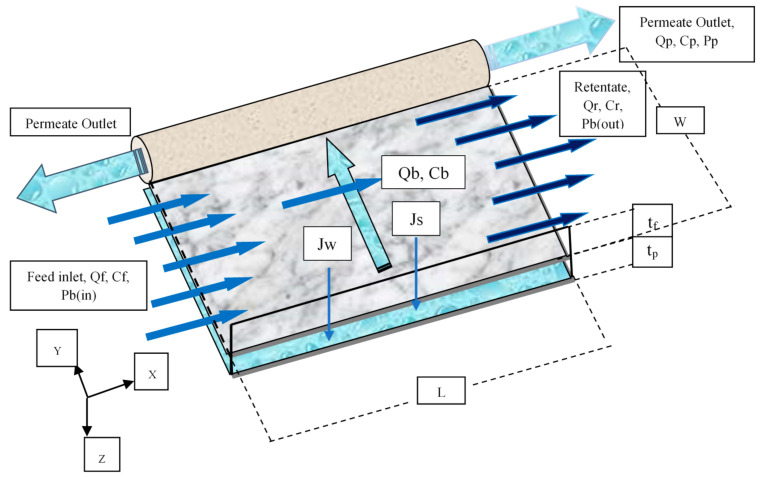
Representation three dimensions’ diagram of a flat sheet membrane (adapted from Al-Obaidi et al. [35]).

**Figure 3 membranes-11-00595-f003:**
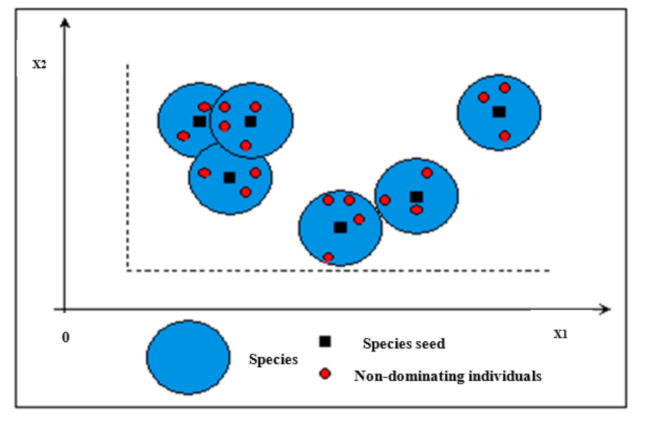
A distribution of species in a 2-D space (adapted from Al-Obaidi et al. [31]).

**Figure 4 membranes-11-00595-f004:**
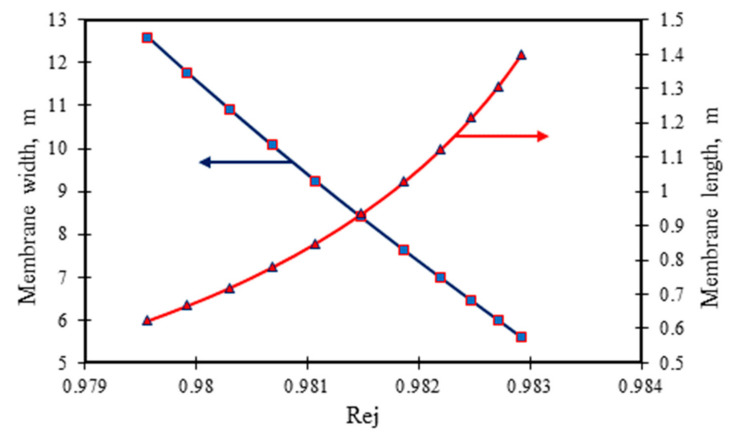
Influence of membrane module design parameters (length and width) on the removal of dimethylphenol at fixed membrane area and feed channel height (operating parameters: (*A*) 6.548 × 10^−3^ kmol/m³, 2.583 × 10^−4^ m³/s, 13.58 atm and 31.5 °C (adapted from Al-Obaidi et al. [24]).

**Figure 5 membranes-11-00595-f005:**
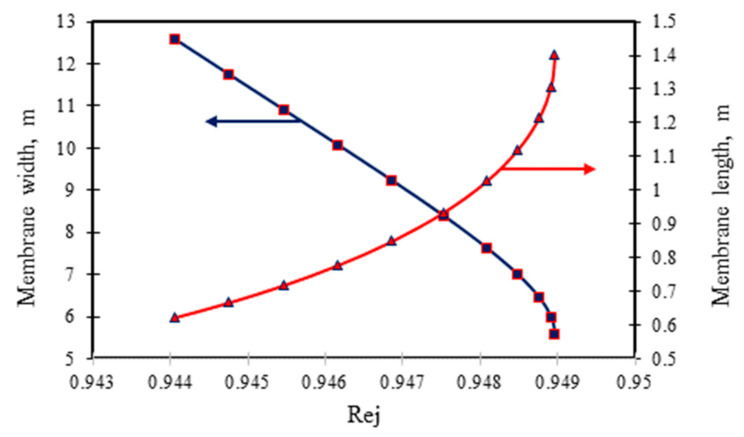
Influence of membrane module design parameters (length and width) on the removal of dimethylphenol at fixed membrane area and feed channel height (operating parameters: (*B*) 0.819 × 10^−3^ kmol/m³, 2.166 × 10^−4^ m³/s, 5.83 atm, and 32.5 °C).

**Figure 6 membranes-11-00595-f006:**
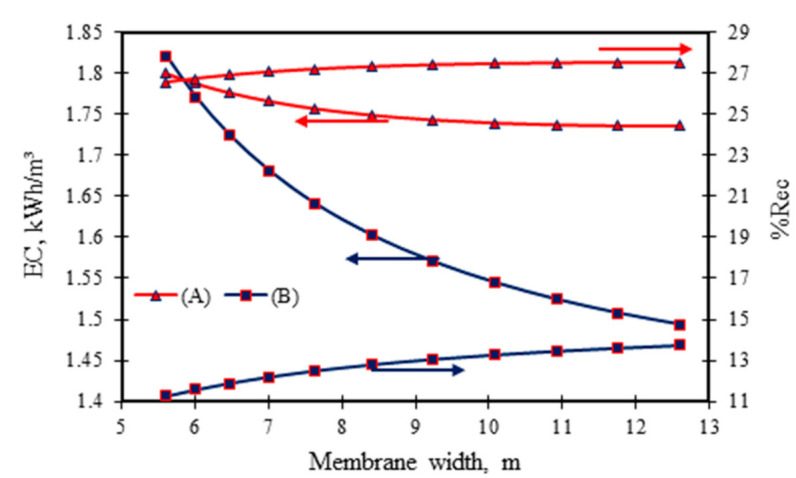
Influence of membrane design width on permeate recovery and energy consumption at fixed membrane area and feed channel height (operating parameters: (*A*) 6.548 × 10^−3^ kmol/m³, 2.583 × 10^−4^ m³/s, 13.58 atm and 31.5 °C, and (*B*) 0.819 × 10^−3^ kmol/m³, 2.166 × 10^−4^ m³/s, 5.83 atm, and 32.5 °C).

**Figure 7 membranes-11-00595-f007:**
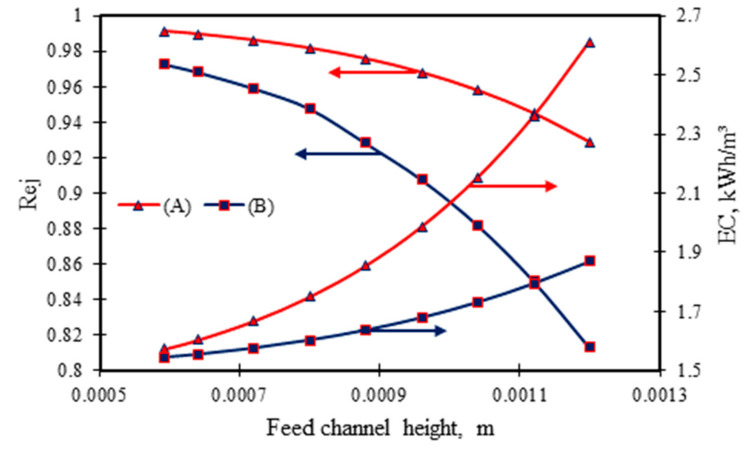
Influence of feed channel height of membrane module on dimethylphenol removal and energy consumption at fixed membrane area (operating parameters: (*A*) 6.548 × 10^−3^ kmol/m³, 2.583 × 10^−4^ m³/s, 13.58 atm and 31.5 °C, and (*B*) 0.819 × 10^−3^ kmol/m³, 2.166 × 10^−4^ m³/s, 5.83 atm, and 32.5 °C).

**Table 1 membranes-11-00595-t001:** Membrane characteristics, dimensions, and transport parameters [33].

Parameter	Ion Exchange, India Ltd. ^1^
Module configuration	Spiral wound membrane
Membrane material	Thin Film Composite Polyamide
Feed (*t*_f_) and permeate (*t*_p_) channel thickness	0.0008 (m) and 0.0005 (m)
Actual membrane area (A)	7.8456 m²
Length (*L*) and width (*W*) of the membrane	0.934 (m) and 8.4 (m)
*b*	9400.9 ((atm s)/m^4^)
*B*_s_ (dimethylphenol)	1.5876 × 10^−8^ (m/s)
*A* _w_	9.7388 × 10^−7^ (m/(atm s))

^1^ Manufacturer.

**Table 2 membranes-11-00595-t002:** Optimisation results of the SCGA.

Control Variables (Case 1)	0.819 × 10^−3^ kmol/m³, 9.71 atm, 2.166 × 10^−4^ m³/s and 32.5 °C)				
Solutions	Variables			Objectives	
	L (m)	W (m)	$t_{f}\;(m)\;\times 10^{3}$	Rej (−)	$EC\;(\mathrm{kWh}/m^{3})$
1	**0.725**	**10.809**	**0.593**	**98.141**	**1.240**
2	0.724	10.822	0.719	96.935	1.264
3	0.728	10.763	0.782	96.096	1.282
**Control Variables** **(Case 2)**	**1.637 × 10^−3^ kmol/m³, 11.64 atm, 2.166 × 10^−4^ m³/s and 31 °C)**				
1	**0.765**	**10.244**	**0.593**	**98.455**	**1.230**
2	0.770	10.177	0.744	97.134	1.286
3	0.803	9.770	0.652	98.055	1.251
**Control Variables** **(Case 3)**	**6.548 × 10^−3^ kmol/m³, 7.77 atm, 2.166 × 10^−4^ m³/s and 31.5 °C)**				
1	**0.756**	**10.374**	**0.593**	**98.400**	**1.528**
2	0.770	10.186	0.626	98.222	1.559
**Control Variables** **(Case 4)**	**6.548 × 10^−3^ kmol/m³, 7.77 atm, 2.33 × 10^−4^ m³/s and 31.5 °C)**				
1	**0.716**	**10.951**	**0.593**	**98.395**	**1.646**
**Control Variables** **(Case 5)**	**2.455 × 10^−3^ kmol/m³, 9.71 atm, 2.583 × 10^−4^ m³/s and 31 °C)**				
1	**0.726**	**10.791**	**0.716**	**97.454**	**1.637**
**Control Variables** **(Case 6)**	**1.637 × 10^−3^ kmol/m³, 11.64 atm, 2.583 × 10^−4^ m³/s and 31 °C)**				
1	**0.822**	**9.534**	**0.677**	**97.894**	**1.520**
Experimental data of Srinivasan et al. [33]	0.934	8.400	0.800	97.300	2.157

## Data Availability

Not applicable.

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
