# Peer review of "Model Based Simulation and Genetic Algorithm Based Optimisation of Spiral Wound Membrane RO Process for Improved Dimethylphenol Rejection from Wastewater"

_membranes, 2021, doi:10.3390/membranes11080595_

Round 1

Reviewer 1 Report

The manuscript deals with "modeling and optimization of dimethylphenol removal by reverse osmosis method".

1. Page 1, Line 32; "insignificant and significant impacts of membrane length and width on dimethylphenol.." Mention the values

2. Page 1, Line 36; "dimethylphenol rejection increased by around 5.8%.." Mention the removal value at the optimum performance?

3. Write keywords alphabetically.

4. Page 4, Line 170; "The model assumptions are as follows:" By reading the paper, I found some other assumptions, which were not listed! For instance, no initial pH difference over running the system!! Premeate flow?!

Without considering the important independent factors in optimization process, the modeling is not fit with real condition!! The current optimization process is just suitable for a lab scale without considering several important independent parameters!

Author Response

Response to Reviewer #1:

The manuscript deals with "modeling and optimization of dimethylphenol removal by reverse osmosis method".

  1. Page 1, Line 32; "insignificant and significant impacts of membrane length and width on dimethylphenol.." Mention the values

Thank you. Although it is plausible to provide the percentages of these impacts in the Abstract, but we think that the important think for this investigation is to judge the type of impact rather than investigating the percentages of this effects. The involvement of these things requires an intensive new paragraph to be included that should provide the variation of both width and length and the associated rejection and energy consumption. This in turn would enlarge the Abstract rather than providing a concise Abstract based on the guidelines of the journal. We have added a new short paragraph in page 13 as highlighted to respond this comment

  1. Page 1, Line 36; "dimethylphenol rejection increased by around 5.8%.." Mention the removal value at the optimum performance?

This is an interesting question raised by the Reviewer. We have added new sentences in page 13 as highlighted to mention the corresponding optimum values of dimethylphenol rejection and energy consumption

For the Abstract, we have added the removal value as requested and highlighted

  1. Write keywords alphabetically.

The keywords are written based on the priority of subject. Therefore, we started with wastewater treatment as the main subject and so on

  1. Page 4, Line 170; "The model assumptions are as follows:" By reading the paper, I found some other assumptions, which were not listed! For instance, no initial pH difference over running the system!! Premeate flow?!

This is another interesting point. The influence of pH variation has not been included in the model. We put it as point (h) as highlighted in page 4.

However, the permeate flow from the membrane pores is already included in Equation 1 in Table A.1 of Appendix A, please see this equation

  1. Without considering the important independent factors in optimization process, the modeling is not fit with real condition!! The current optimization process is just suitable for a lab scale without considering several important independent parameters!

We totally agree with the statement above. However, the authors were already published intensive studies regarding the influence of control variables of the process (operating conditions) on the process performance throughout intensive optimisation. We think that filling the gap in the literature is important for this stage of research since there was no study has investigated the influence of dimensions of a membrane on the performance indicators in a thorough simulation and optimisation using Species Conserving Genetic Algorithm method   

Reviewer 2 Report

The title is clear.

The content is in accord with title.

The manuscript adheres to the journal's standards after revision.

The size of the article is appropriate to the contents.

The Abstract must be revised. The Abstract must refer to the study findings, methodologies, discussion as well as conclusion.

The key words permit found article in the current registers or indexes.

In the introduction it is clearly described the state of the art of the investigated problem. The references from the 2021 are necessary.

The methodology is adequately described.

The figures have a good quality. Figure 3 is not clear for readers.

The tables contain necessary results.

The Conclusion is OK.

References from 2021 year are necessary.

The paper has the text presented and arranged clearly and concisely.

Please respect guide for authors. The References aren’t in format.

Author Response

Response to Reviewer #2:

The title is clear.

The content is in accord with title.

The manuscript adheres to the journal's standards after revision.

The size of the article is appropriate to the contents.

  1. The Abstract must be revised. The Abstract must refer to the study findings, methodologies, discussion as well as conclusion.

Thank you. We have revised the Abstract to reflect our response to the Reviewer’s comment above. The methodology and most important findings are highlighted for the convenience of the Reviewer

  1. The key words permit found article in the current registers or indexes.

Thank you

  1. In the introduction it is clearly described the state of the art of the investigated problem. The references from the 2021 are necessary.

We have already added some references of recent published papers (2020) as highlighted in the Introduction. However, to respond this comment, we have added new two references for published papers in 2021. Please see the highlighted sentences in the Introduction and the reference list

  1. The methodology is adequately described.

Thank you

  1. The figures have a good quality. Figure 3 is not clear for readers.

Thank you. We have edited Figure 3 as requested. Enlarging the size of the figure was the reason of getting a bit disruption. The size is refined now

  1. The tables contain necessary results.

Thank you

  1. The Conclusion is OK.

Thank you

  1. References from 2021 year are necessary.

We have added new references of published papers in 2021 as highlighted

  1. The paper has the text presented and arranged clearly and concisely.

Thank you

  1. Please respect guide for authors. The References aren’t in format.

We have already traced the guidelines of writing the references

Round 2

Reviewer 1 Report

I still believe with ignoring several important parameters during running the modeling and optimization, this paper is not suitable for future studies. Anyway, I pass the final decision about it to the Editor.

The rest comments have been addressed.